# Investigation of Novel Predictive Biomarkers for Preeclampsia in Second-Trimester Amniotic Fluid

**DOI:** 10.3390/ijms262110530

**Published:** 2025-10-29

**Authors:** Hyo Eun Lee, Yeonseong Jeong, Jue Young Kim, Ha-Yeon Shin, Young-Han Kim, Min-A Kim

**Affiliations:** 1The Graduate School, Yonsei University College of Medicine, Seoul 03722, Republic of Korea; cinthia79@naver.com; 2Department of Obstetrics and Gynecology, Gangnam Severance Hospital, Yonsei University College of Medicine, Seoul 06273, Republic of Korea; 928sophia928@gmail.com (Y.J.); forsythia410@yuhs.ac (J.Y.K.); hayeon37@yuhs.ac (H.-Y.S.); 3Department of Obstetrics and Gynecology, Severance Hospital, Yonsei University College of Medicine, Seoul 06273, Republic of Korea; yhkim522@yuhs.ac

**Keywords:** preeclampsia, sequencing analysis, predictive markers, amniotic fluid

## Abstract

Preeclampsia (PE) is a major cause of maternal and perinatal morbidity, and early prediction is critical for timely intervention. This study aimed to identify predictive biomarkers for PE through transcriptomic analysis of second-trimester amniotic fluid supernatant (AFS) collected prior to clinical symptom onset. AFS samples from women who later developed PE (n = 7) and matched controls (n = 7) underwent RNA sequencing to identify differentially expressed genes (DEGs). Candidate genes were validated by real-time PCR in HTR-8/SVneo cells exposed to fluid shear stress at 3, 10, and 20 dyn/cm^2^ for 24 h, mimicking the hemodynamic environment of PE, and siRNA-mediated knockdown was used to assess effects on trophoblast migration and invasion. RNA sequencing revealed 19 DEGs, with 3 upregulated and 16 downregulated genes in the PE group. HOOK2 emerged as the most significantly upregulated gene. Four candidate genes, including HOOK2, CCDC160, CKB, and PARP15, were selected for further validation. HOOK2 mRNA expression significantly increased with higher shear stress levels, consistent with sequencing data. Knockdown of HOOK2 led to a significant increase in trophoblast invasion, while migration showed no significant change. These findings suggest that HOOK2 may serve as a promising early biomarker for PE by modulating trophoblast invasiveness under altered hemodynamic conditions, with potential to improve risk stratification and personalized monitoring in pregnancy.

## 1. Introduction

Preeclampsia (PE) is a pregnancy-specific hypertensive disorder that affects 2–8% of all pregnancies [1]. The clinical spectrum of PE ranges from mild hypertension and proteinuria to serious manifestations, including headache, visual disturbances, and epigastric pain. In its severe form, PE may progress to multiorgan dysfunction involving the liver, kidneys, lungs, and brain [2]. A consistent upward trend in maternal age has been observed in recent years, correlating with increased severity of PE [3]. As delivery remains the only definitive treatment, early diagnosis and optimal timing of delivery are crucial to improving maternal and fetal outcomes [4].

Early prediction of PE would allow preventive strategies and timely interventions that could reduce maternal morbidity and minimize the risk of preterm delivery. The PHOENIX trial demonstrated that planned early delivery for late-preterm PE significantly reduced maternal complications [5]. Consequently, research has focused on developing biomarkers for earlier detection of PE [6,7]. Several proteins, such as placental growth factor (PlGF), soluble fms-like tyrosine kinase-1 (sFlt-1), soluble endoglin (sEng), and pregnancy-associated plasma protein A (PP13), have been proposed as potential biomarkers. However, these markers usually become detectable only after disease progression, and variability in immunoassay performance further limits their diagnostic utility [8,9]. Thus, reliable biomarkers for early prediction remain an unmet need.

Amniotic fluid (AF) is essential for fetal development and intrauterine homeostasis, and it has long been used as a diagnostic medium for assessing fetal lung maturity, intrauterine infection, and chromosomal abnormalities [10,11]. Typically, only cells extracted from AF are utilized in these diagnostic methods, with the remainder of the AF supernatant (AFS) often discarded. Notably, AFS contains cell-free nucleic acids (cfNAs), including DNA and RNA, which originate from cellular turnover and biological processes in the placenta [12,13,14]. However, the potential of AFS-derived cfNAs as a source of biomarkers for PE has been largely unexplored.

In this study, RNA sequencing was used to profile differential gene expression in AFS collected prior to the clinical onset of PE. Candidate transcripts were validated by reverse transcription polymerase chain reaction (RT-PCR) in human trophoblast cells exposed to fluid shear stress, which mimics the hemodynamic alterations characteristic of PE. In particular, chorionic trophoblasts in PE are exposed to abnormally high levels of shear stress, contributing to impaired placental development and disease progression, which justifies the use of shear stress experiments in this study. This approach provides a physiologically relevant model for validating molecular changes identified in AFS.

Currently, serum biomarkers remain the primary tools under investigation for PE screening. However, their limited sensitivity, specificity, and late detectability underscore the need for novel approaches [15]. We hypothesized that AFS-derived transcripts could serve as earlier and more reliable biomarkers for PE. By investigating AFS as an alternative molecular source, this study aims to identify novel gene expression signatures with the potential to improve early prediction and clinical management of PE.

## 2. Results

### 2.1. Clinical Characteristics of the Study Population

The demographic and clinical characteristics of the study population are summarized in Table 1. There were no significant differences in maternal age (35.1 ± 2.7 vs. 37.6 ± 2.5 years) or gestational age at amniocentesis (17.8 ± 1.4 vs. 17.7 ± 1.0 weeks) between the PE and control groups. In the PE group, the gestational age at diagnosis ranged from 29 + 4 to 39 + 1 weeks (median, 37 + 0 weeks), indicating a spectrum from early- to late-onset PE. Indications for amniocentesis included advanced maternal age, positive maternal serum screening tests for Down syndrome or neural tube defects, increased nuchal translucency, and abnormal ultrasound findings. However, the final amniocentesis results confirmed that all included cases had normal chromosomal findings.

### 2.2. HOOK2 as the Most Overexpressed Gene in Preeclampsia

Figure 1A illustrates the analytical workflow applied to the 14 AFS samples from the PE and control groups. Differential gene expression was defined as a fold change (FC) > 1.5, *p* < 0.05, and false discovery rate (FDR) < 0.05. To identify robust differentially expressed genes (DEGs), two analytical approaches were compared: a *t*-test and Cuffdiff analysis. The *t*-test yielded 534 DEGs (FC > 1.5, *p* < 0.05), whereas Cuffdiff identified 37 DEGs (FC > 1.5, *p* < 0.05, FDR < 0.05). Nineteen genes overlapped between the two methods, representing consistently DEGs associated with PE. Figure 1B presents a heatmap of these 19 overlapping DEGs in second-trimester AFS samples, demonstrating distinct expression profiles between the PE and control groups. Among these, HOOK2, CCDC160, and CKB were significantly upregulated, while 16 genes were downregulated in the PE group. The representative DEGs with the largest fold changes are summarized in Table 2. HOOK2 showed the highest upregulation (FC = 3.899, *p* = 0.006, FDR = 0.031), followed by CCDC160 (FC = 3.444), CKB (FC = 3.413), and PARP15 (FC = –1.179).

### 2.3. HOOK2 Expression Levels Were Increased in HTR-8/SVneo Cells by Fluid Shear Stress

To simulate PE conditions, HTR-8/SVneo cells were cultured under fluid shear stress using a pump system. Cells were exposed for 24 h to shear stress intensities of 3, 10, or 20 dyn/cm^2^ to examine changes in candidate gene expression. Shear stress of 3 dyn/cm^2^ was used as the control, as the average shear stress experienced by chorionic villi is <5 dyn/cm^2^ [16,17,18].

To validate whether the in vitro experiments accurately replicated the placental perfusion environment, we assessed the mRNA expression of well-known markers for PE and trophoblasts (Figure 2A). Real-time qPCR showed that the mRNA expression of FLT-1 and CGB3 increased compared to the control conditions as higher fluid shear stresses were applied. PlGF mRNA expression increased at 10 dyn/cm^2^ compared with control, but was reduced at 20 dyn/cm^2^ relative to the 10 dyn/cm^2^ condition. While fluid shear stress promotes PlGF expression in human trophoblasts [18], excessive levels of shear stress under pathological conditions such as PE may instead suppress its expression.

Considering the central role of oxidative stress in the pathophysiology of PE, we investigated oxidative stress markers after exposure to fluid shear stress, as illustrated in Figure 2B. Consistent with previous reports, the expression of GPX1 decreased following fluid shear stress, whereas SOD2 expression increased. Subsequently, we selected candidate genes based on the differential expression analysis from RNA sequencing results, as depicted in Figure 2C. We chose HOOK2, CCDC160, CKB, and PARP15, which emerged as DEGs from the RNA sequencing data. These genes were further examined for mRNA expression differences using RT-PCR in HTR-8/SVneo cells exposed to shear stress conditions of 3, 10, and 20 dyn/cm^2^ for 24 h. HOOK2 mRNA levels were found to increase significantly with escalating shear stress, showing marked elevation from the control group to 10 and 20 dyn/cm^2^, respectively. This pattern of HOOK2 upregulation was in accordance with the RNA sequencing findings. Notably, exposure to higher shear stress intensities resulted in statistically significant increases in HOOK2 expression (10 dyn/cm^2^: *p* < 0.0001; 20 dyn/cm^2^: *p* < 0.001).

### 2.4. Reducing HOOK2 Level Increased Invasiveness of HTR-8/SVneo Cells

To evaluate the impact of HOOK2 on motility in HTR-8/SVneo cells, HOOK2 expression was silenced using siRNA, with si-control used as a comparative control (Figure 3A). No discernible difference in cell proliferation was observed with siRNA treatment (Figure 3B). While GPX1 expression exhibited a decrease following HOOK2 knockdown, this decrease was not statistically significant (Figure 3C). In contrast, SOD2 expression significantly decreased after HOOK2 depletion (Figure 3D).

The siRNA treatment effectively suppressed HOOK2 expression, with higher siRNA concentrations resulting in greater inhibition. To investigate the effect of HOOK2 knockdown on the migratory and invasive capacities of HTR-8/SVneo cells, migration and invasion assays were performed. Although migration activity showed a decrease following HOOK2 knockdown, this reduction was not statistically significant (Figure 3E). In contrast, invasion activity was significantly increased after HOOK2 depletion compared to the si-control group (Figure 3F). These findings suggest a potential role for HOOK2 in regulating the invasion of HTR-8/SVneo cells.

## 3. Discussion

The present study aimed to identify differential gene expression in second-trimester AFS prior to the clinical onset of PE. Among the candidate genes identified through RNA sequencing, HOOK2 emerged as the most significantly upregulated. To validate these findings, human trophoblast cells (HTR-8/SVneo) were exposed to varying levels of fluid shear stress, mimicking the hemodynamic alterations characteristic of PE. Under these conditions, HOOK2 expression increased in a dose-dependent manner, consistent with the RNA sequencing results, supporting a potential role in the pathophysiology of PE. Functional assays revealed that HOOK2 knockdown enhanced trophoblast invasion, whereas migration was not significantly affected.

These observations suggest that HOOK2 may regulate invasive capacity under pathophysiological shear stress, potentially influencing spiral artery remodeling. Spiral artery remodeling is essential for normal placental development, as trophoblast invasion into the vascular media leads to vessel dilation and improved maternal-fetal perfusion [19,20]. Impaired remodeling is a hallmark of PE, contributing to adverse maternal and fetal outcomes [21]. Fluid shear stress refers to the frictional force generated by blood flow acting parallel to the endothelial surface. In normal pregnancies, these hemodynamic forces support the physiological remodeling of spiral arteries [22]. In contrast, altered shear stress in PE contributes to endothelial dysfunction and may exacerbate shallow trophoblast invasion and inadequate spiral artery remodeling [17]. In our in vitro model, increasing shear stress altered the expression of HOOK2 as well as PE-related markers (FLT-1, PlGF, CGB3), reflecting pathophysiologically relevant responses.

HOOK2 is a member of the Hook protein family, functioning as a cytoskeletal linker that connects microtubules to subcellular structures and regulates centrosome organization, vesicle transport, and polarized cell migration [23,24,25], although its biological roles remain comparatively less well characterized. It interacts with centrosomal proteins and polarity regulators, including the PAR6α/PAR3/aPKC complex, ensuring proper centrosome and Golgi orientation. Beyond these roles, HOOK2 contributes to pericentrosomal aggresome formation—structures enriched in misfolded proteins, chaperones, and proteasomes—by mediating retrograde microtubule transport of protein aggregates to the centrosome [24].

Excessive HOOK2 expression, as observed in this study, may lead to abnormal pericentrosomal clustering of polarity regulators, potentially stabilizing microtubule–centrosome interactions excessively and reducing cytoskeletal flexibility [26]. A potential consequence of such dysregulation is impaired ciliogenesis. Primary cilia act as mechanosensors in trophoblasts, allowing cells to respond to hemodynamic forces and coordinate spiral artery remodeling [27]. Previous studies in neuronal and epithelial cells have shown that pericentrosomal aggresomes disrupt centrosomal function and ciliary assembly [28]. By analogy, HOOK2 overexpression may impede ciliogenesis in trophoblasts, reducing the ability to sense shear stress and consequently limiting invasive capacity. This mechanistic hypothesis aligns with our finding that HOOK2 knockdown enhanced invasion, suggesting that overexpression may restrict trophoblast invasiveness through altered polarity signaling and centrosome/Golgi orientation.

Oxidative stress is another critical factor in PE pathophysiology. In HTR-8/SVneo cells, HOOK2 knockdown led to a decrease in SOD2 expression, whereas GPX1 reduction was not statistically significant. These findings suggest a potential link between HOOK2 expression and the cellular oxidative stress response, though the exact molecular mechanisms require further elucidation.

The potential of HOOK2 as an early biomarker for PE is notable, as it is detectable prior to clinical manifestation. AFS represents a rich source of placental-derived molecular material, including cell-free nucleic acids and extracellular vesicles, providing a window into early placental pathophysiology [12]. Unlike maternal blood-based biomarkers, which may be diluted systemically, AFS allows detection of subtle molecular alterations preceding PE. However, AFS collection is limited to clinically indicated amniocentesis, constraining routine clinical use. Thus, while HOOK2 shows promise as an early biomarker, translation to broader practice may require validation in maternal blood or other less invasive samples.

Beyond HOOK2, our transcriptomic analysis also identified 19 DEGs, including CCDC160, CKB, and PARP15, which were also validated at the transcriptional level. However, the functional interconnections and biological significance of these genes in PE remain insufficiently characterized, warranting further investigation. Among these additional candidates detected in this study, CCDC160 encodes a coiled-coil domain-containing protein potentially linked to cytoskeletal organization and vesicular trafficking, pathways also relevant to trophoblast motility and endosomal regulation [29]. CKB participates in intracellular ATP buffering and metabolic homeostasis and has been implicated in oxidative-stress-induced trophoblastic dysfunction and aberrant energy metabolism in PE placentas [30]. PARP15, a poly (ADP-ribose) polymerase family member, is involved in DNA repair and post-translational regulation of stress responses, and this family has been increasingly associated with trophoblast inflammation and ferroptotic signaling in PE [31]. Future network-based and functional analyses are needed to determine whether these DEGs act in coordinated pathways influencing trophoblast function and PE pathogenesis.

It is important to recognize that, while this study focused exclusively on the effects of fluid shear stress on trophoblast function, PE is a multifactorial disorder with physiological complexity that extends beyond the scope of the in vitro model employed. In vivo, trophoblasts are exposed to a broad range of mechanical, oxidative, inflammatory, and hypoxic stimuli, each of which can independently or synergistically influence placental gene expression and cellular behavior [32,33,34]. Therefore, while the fluid shear stress-based in vitro model provides significant mechanobiological insights, it only partially replicates the complex microenvironment of the preeclamptic placenta. Future investigations should integrate multiple pathophysiological stressors—including hypoxia and inflammatory mediators—to achieve a more physiologically comprehensive understanding of trophoblast dysfunction in PE.

Several limitations should be acknowledged. First, the small sample size limits statistical power and generalizability. Nonetheless, these prospectively collected second-trimester amniotic fluid samples—obtained before the clinical onset of PE—represent a rare and valuable resource for early biomarker discovery. Given the low prevalence of PE, future studies should validate these candidate biomarkers in larger, independent cohorts and across diverse biospecimens (e.g., maternal plasma or placental tissue) to enhance generalizability and translational relevance. Second, mechanistic interpretations regarding centrosome polarity, ciliogenesis, and cytoskeletal regulation are based on literature extrapolation rather than direct experimental measurement. The study also could not assess PE subtypes or longitudinal HOOK2 expression dynamics across gestation. Future studies should address these gaps by including larger, diverse cohorts, performing in vivo validation, and incorporating multi-time-point analyses. Moreover, combining HOOK2 with established angiogenic markers (sFlt-1, PlGF) may enhance predictive performance.

Finally, the functional validation experiments were conducted using the immortalized HTR-8/SVneo trophoblast cell line. While this cell line is widely employed for investigating trophoblast migration, invasion, and response to shear stress because of its reproducibility and ease of culture, it may not fully reflect the molecular diversity or invasive behavior of primary trophoblasts in vivo [35]. Therefore, although our findings in HTR-8/SVneo cells provide valuable mechanistic insights, future studies employing primary trophoblasts, placental explant cultures, or trophoblast organoid models are warranted to confirm the role of HOOK2 under physiologically relevant conditions.

In summary, HOOK2 is upregulated in second-trimester AFS prior to PE onset and modulates trophoblast invasion under shear stress. These findings support HOOK2 as a candidate early biomarker and provide mechanistic insight into its potential role in PE pathogenesis. Further studies are warranted to validate these results, elucidate molecular mechanisms, and assess the utility of HOOK2 in combination with other biomarkers for early detection and risk stratification in PE.

## 4. Materials and Methods

### 4.1. Patients and Amniotic Fluid Collection

AFS samples were obtained from 14 asymptomatic pregnant women who underwent amniocentesis for genetic diagnosis between March 2011 and May 2017 at Gangnam Severance Hospital, Seoul, Korea. Samples were collected at 16–21 weeks of gestation, comprising 7 women who subsequently developed PE and 7 unaffected controls. Exclusion criteria included preterm labor, multiple gestation, preexisting chronic hypertension, pregestational diabetes, autoimmune diseases, fetal structural anomalies or chromosomal abnormalities, clinical chorioamnionitis or maternal infection, and significant maternal comorbidities. Total RNA was extracted from the collected AFS, and samples were categorized according to the subsequent development of PE. Total RNA was isolated from 300 μL of AFS using Trizol reagent (Cat. No. 15596026, Invitrogen, Carlsbad, CA, USA) according to the manufacturer’s protocol. RNA quality was assessed by Agilent 2100 Bioanalyzer (Agilent Technologies, Santa Clara, CA, USA), and RNA quantification was performed using an ND-2000 Spectrophotometer (Thermo Fisher Scientific, Waltham, MA, USA). All procedures were approved by the Institutional Review Board of Yonsei University College of Medicine, Seoul, Republic of Korea (IRB #3-2024-0427) and were conducted in accordance with relevant guidelines and regulations. Written informed consent was obtained from all participants prior to study enrollment and amniocentesis.

### 4.2. RNA Sequencing and Analysis of Differentially Expressed Genes

NEBNext Ultra II Directional RNA-Seq Kit (E7760, New England Biolabs, Ipswich, MA, USA) was utilized to generate libraries from 100 ng–1 μg of total RNA. High-throughput sequencing was conducted using the NovaSeq 6000 platform (Illumina, San Diego, CA, USA) with paired-end 100 bp reads. Raw sequencing data underwent quality control assessment employing FastQC v0.11.9 (Babraham Bioinformatics, Cambridge, UK) [36], and Read Count (RC) data were processed in EdgeR [37] within R [38] following the FPKM+Geometric normalization method. Differential expression analysis between the PE and control groups was executed, and genes exhibiting differential expression were pinpointed as prospective biomarkers of PE if they met the following criteria: FC > 1.5, *p*-value < 0.05, and FDR < 0.05 using the Cuffdiff program [39,40]. Significant differences in gene expression were determined via Student’s *t*-test. DEGs were selected if they were detected by both methods. The dataset of identified potential biomarkers was imported into the MeV software package (v4.9, The Institute for Genomic Research, Rockville, MD, USA) for heatmap generation and hierarchical cluster analysis (HCA).

### 4.3. Cell Culture

The human trophoblast cell line HTR-8/SVneo was obtained from the American Type Culture Collection (ATCC, Manassas, VA, USA) and cultured in RPMI 1640 medium (Cat. No. 10-040-CV, Corning, Corning, NY, USA) supplemented with 5% fetal bovine serum (FBS; Cat. No. 12483-020, Gibco, Grand Island, NY, USA) and 1% penicillin/streptomycin (Cat. No. 15140-122, Gibco, Grand Island, NY, USA). Cultures were maintained in a humidified incubator at 37 °C with 5% CO_2_, and the culture medium was replenished every 3 days.

### 4.4. Fluid Shear Stress Experiments

The HTR-8/SVneo cells were seeded at a density of 5 × 10^5^ cells/mL in μ-Slide I0.4 Luer chambers (Cat. No. 80171, Ibidi, Gräfelfing, Germany) and allowed to adhere for 24 h. Following this, the cells were exposed to shear stress conditions (3, 10, or 20 dyn/cm^2^) for 24 h utilizing the ibidi Pump System (Cat. No. 10902, Ibidi, Gräfelfing, Germany).

### 4.5. RNA Extraction and Real-Time PCR

Total RNA extraction from each cell was accomplished using the RNeasy Micro Kit (Cat. No. 74004, Qiagen, Hilden, Germany), and cDNA synthesis was carried out using 0.5 µg of total RNA with the Maxima First Strand cDNA Synthesis Kit (Cat. No. K1641, Thermo Scientific, Waltham, MA, USA) following the respective manufacturers’ protocols. Real-time PCR was performed to quantify mRNA expression utilizing Fast SYBR™ Green Master Mix (Cat. No. 4385612, Applied Biosystems, Foster City, CA, USA) and a StepOnePlusTM Real-Time PCR system (Applied Biosystems, Foster City, CA, USA), as per the manufacturer’s instructions. The primer sequences for PCR are detailed in Table 3. Results were normalized to β-actin expression.

### 4.6. siRNA Transfection

HOOK2-specific and negative control siRNAs were purchased from Bioneer Corporation (Daejeon, Republic of Korea). siRNA transfection was performed using Lipofectamine RNAiMax (Cat. No. 13778-150, Invitrogen, Carlsbad, CA, USA) according to the manufacturer’s protocol. Transfection efficiency was assessed 48 h after transfection.

### 4.7. Transwell Migration, Invasion Assay

For the assessment of cell migration and invasion, 24-well transwell plates equipped with 8-μm pore inserts (Cat. No. 353097, Corning Life Sciences, Corning, NY, USA) were employed. For the invasion assay, the insert membrane was coated with Matrigel (50 μL/well; Cat. No. 354234, BD Biosciences, San Jose, CA, USA) and allowed to solidify for 24 h. Cells (2 × 10^4^) in serum-free medium were seeded in the upper inserts, while culture medium containing 10% FBS was added to the lower wells. Following 48 h of incubation, cells that had migrated or invaded through the insert pores were fixed, stained with 0.25% crystal violet, and visualized under light microscopy (Axio Imager.M2, Carl Zeiss, Oberkochen, Germany). Cell counts were performed in three random fields at a magnification of 200×.

### 4.8. Statistical Analyses

All statistical analyses were performed using SPSS version 29.0 (IBM Corp., Armonk, NY, USA). In vitro data were obtained from three independent experiments, each conducted at least in triplicate for all variables. To evaluate differences between the control and experimental groups, Student’s *t*-tests or Mann–Whitney U tests were utilized. A significance level of *p*-value < 0.05 was considered statistically significant.

## Figures and Tables

**Figure 1 ijms-26-10530-f001:**
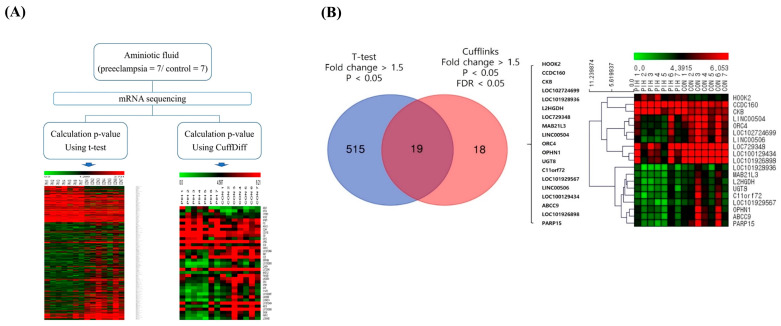
Workflow and identification of differentially expressed genes (DEGs) in preeclampsia. (**A**) Flow chart summarizing the two analysis pipelines applied to 14 amniotic fluid supernatant samples from preeclampsia and control groups, accompanied by a heatmap illustrating the DEGs identified by each analysis method. (**B**) Overlap analysis and heatmap of DEGs identified by both *t*-test and Cufflinks methods.

**Figure 2 ijms-26-10530-f002:**
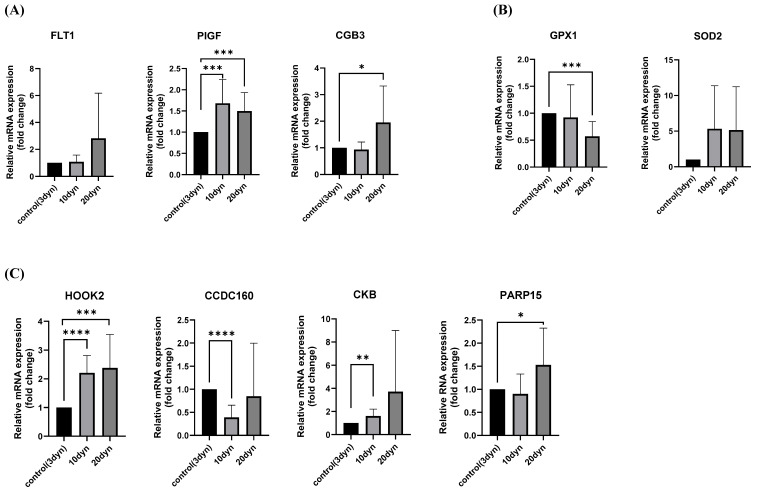
Validation of preeclampsia (PE)-associated gene expression in trophoblasts exposed to fluid shear stress. (**A**) Expression of established PE markers (FLT-1, PlGF, CGB3) in HTR-8/SVneo cells under increasing shear stress. (**B**) Expression of oxidative stress markers (GPX1, SOD2) in response to shear stress. (**C**) Expression of candidate DEGs (HOOK2, CCDC160, CKB, PARP15) in HTR-8/SVneo cells exposed to shear stress (* *p* < 0.05, ** *p* < 0.01, *** *p* < 0.001, **** *p* < 0.0001).

**Figure 3 ijms-26-10530-f003:**
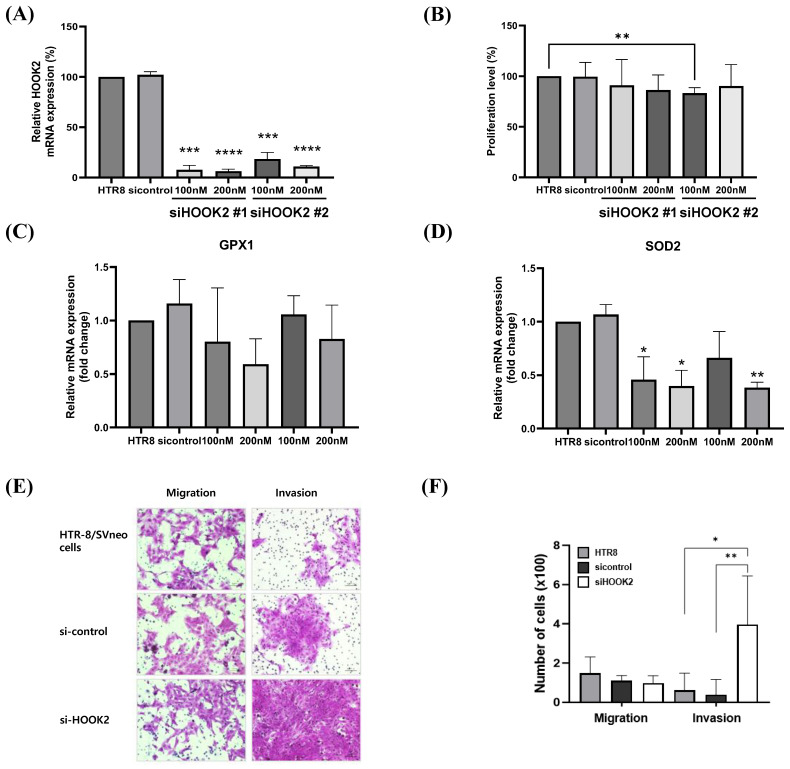
Functional analysis of HOOK2 knockdown in trophoblasts. (**A**) HOOK2 expression in HTR-8/SVneo cells following siRNA-mediated knockdown compared with si-control (**B**) Cell proliferation in HTR-8/SVneo cells following HOOK2 knockdown. (**C**,**D**) Expression of oxidative stress markers GPX1 (**C**) and SOD2 (**D**) after HOOK2 knockdown. (**E**,**F**) Effect of HOOK2 knockdown on trophoblast motility: representative microscopic images (**E**) and quantitative analysis of migration and invasion (**F**) (* *p* < 0.05, ** *p* < 0.01, *** *p* < 0.001, **** *p* < 0.0001).

**Table 1 ijms-26-10530-t001:** Clinical characteristics of the study population.

Sample	Age (Years)	Gestational Age at Amniocentesis (Weeks)	Indication of Amniocentesis
P1	40	17 + 3	AMA
P2	32	17 + 2	Positive screening test for neural tube defect
P3	33	17 + 0	Positive screening test for Down syndrome
P4	34	17 + 3	Nuchal translucency dilatation
P5	39	16 + 3	AMA
P6	32	21 + 3	Abnormal ultrasonography finding
P7	36	17 + 2	AMA
C1	39	16 + 4	Positive screening test for Down syndrome
C2	38	19 + 6	AMA
C3	35	17 + 1	AMA
C4	40	16 + 1	AMA
C5	36	18 + 2	Positive screening test for Down syndrome
C6	33	18 + 0	Positive screening test for Down syndrome
C7	42	17 + 6	AMA

P, Preeclampsia group; C, Control group; AMA, Advanced maternal age.

**Table 2 ijms-26-10530-t002:** Representative differentially expressed genes between preeclampsia and control groups.

Gene Symbol	Fold Change	*p*-Value	False Discovery Rate
	PE/CON	PE/CON	PE/CON
HOOK2	3.899	0.006	0.031
CCDC160	3.444	0.015	0.031
CKB	3.413	0.023	0.031
PARP15	−1.179	0.042	0.031

PE, Preeclampsia group; CON, control group.

**Table 3 ijms-26-10530-t003:** Primer sequences used for real-time PCR.

Genes	Sense	Antisense
FLT1	AGAGGTGAGCACTGCAACAA	TCTCCTCCGAGCCTGAAAGT
PlGF	ACCTGATGGTAGGAGGCAGT	ACCAGAACAGATGCACAACCA
CGB3	CACCCCAGCATCCTATCACC	ATCTCCATCCTTGGTGCGTC
GPX-1	TATCGAGAATGTGGCGTCCC	TCTTGGCGTTCTCCTGATGC
SOD2	GCTGGAAGCCATCAAACGTG	GCCTGTTGTTCCTTGCAGTG
HOOK2	CCAACTGGAAGCTGAAGGTC	GTCCTGCTTTTTCTCGCAAC
CCDC160	GCAAGCCAAAGAAGTCATCC	GCGGATCTTTGCCATTTCTA
CKB	TTCTCAGAGGTGGAGCTGGT	TACCAAGGGTGACGGAAGTC
PARP15	TGGGACAGATGCAGACTCAG	GCTGTCTGGCTTGGAGTAGG
β-actin	GGACTTCGAGCAAGAGATGG	AGCACTGTGTTGGCGTACAG

## Data Availability

The original contributions presented in this study are included in the article. Further inquiries can be directed to the corresponding authors.

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
