# Peer review of "Investigation of Novel Predictive Biomarkers for Preeclampsia in Second-Trimester Amniotic Fluid"

_ijms, 2025, doi:10.3390/ijms262110530_

Round 1

Reviewer 1 Report

Comments and Suggestions for Authors
  1. The structure of the manuscript is not standard.

Move 4. Materials and Methods line227-305 to line 73.

  1. For PEgroup, please detail the gestational week of diagnosis
  2. Please elucidate the genes expression in two groups in the results, it’s not clear.
  3. Add in the limitations, only consider the fluid shear stress, however, in PE, there are many influence factors.

Author Response

Response to Reviewers

Reviewer #1

Comments 1: The structure of the manuscript is not standard. Move 4. Materials and Methods line227-305 to line 73.

Response 1: We sincerely appreciate the reviewer’s thoughtful comment regarding the manuscript structure. We fully respect the reviewer’s concern about maintaining a clear and logical organization. However, according to the official Instructions for Authors of the International Journal of Molecular Sciences (IJMS), the journal adopts the IMRaD structure, in which the Materials and Methods section appears after the Discussion. Therefore, to comply with the journal’s official formatting standards, we have maintained the current structure. We greatly value the reviewer’s input and have carefully rechecked the section headings to ensure consistency and clarity throughout the manuscript. If this structure is considered inappropriate by the editor or reviewers, we are willing to revise it accordingly.

Comments 2: For PE group, please detail the gestational week of diagnosis.

Response 2:  We thank the reviewer for this valuable suggestion. The gestational ages at diagnosis for the seven women who developed PE were 39+0, 39+1, 38+6, 37+0, 34+6, 29+4, and 31+4 weeks, respectively. This information has been added to the results section to clarify the clinical characteristics of the PE group. The revised sentence reads as follows:

“In the PE group, the gestational age at diagnosis ranged from 29+4 to 39+1 weeks (median, 37+0 weeks), indicating a spectrum from early- to late-onset PE.”

Comments 3: Please elucidate the genes expression in two groups in the results, it’s not clear.

Response 3: We appreciate the reviewer’s valuable comment. To address this, we have added the following description to the results section to more clearly present the differential gene expression patterns between the PE and control groups.

“Figure 1A illustrates the analytical workflow applied to the 14 AFS samples from the PE and control groups. Differential gene expression was defined as a fold change (FC) > 1.5, P < 0.05, and false discovery rate (FDR) < 0.05. To identify robust differentially expressed genes (DEGs), two analytical approaches were compared: a T-test and Cuffdiff analysis. The T-test yielded 534 DEGs (FC > 1.5, P < 0.05), whereas Cuffdiff identified 37 DEGs (FC > 1.5, P < 0.05, FDR < 0.05). Nineteen genes overlapped between the two methods, representing consistently DEGs associated with PE. Figure 1B presents a heatmap of these 19 overlapping DEGs in second-trimester AFS samples, demonstrating distinct expression profiles between the PE and control groups. Among these, HOOK2, CCDC160, and CKB were significantly upregulated, while 16 genes were downregulated in the PE group. The representative DEGs with the largest fold changes are summarized in Table 2. HOOK2 showed the highest upregulation (FC = 3.899, P = 0.006, FDR = 0.031), followed by CCDC160 (FC = 3.444), CKB (FC = 3.413), and PARP15 (FC = –1.179).”

Comments 4: Add in the limitations, only consider the fluid shear stress, however, in PE, there are many influence factors.

Response 4: We appreciate the reviewer’s valuable comment. We fully agree that PE is a multifactorial disorder, and our study focused on fluid shear stress as one hemodynamic factor influencing trophoblast behavior. To address this concern, we have added a statement in the limitations section emphasizing that PE pathophysiology involves complex interactions among hemodynamic, immune, angiogenic, and oxidative mechanisms. While our shear stress model provides a physiologically relevant approach to study mechanical stimuli, it does not capture the full spectrum of molecular and cellular perturbations occurring in vivo. We have also cited relevant literature supporting the multifactorial nature of PE. Therefore, we have added the following statement to the discussion section:

“It is important to recognize that, while this study focused exclusively on the effects of fluid shear stress on trophoblast function, PE is a multifactorial disorder with physiological complexity that extends beyond the scope of the in vitro model employed. In vivo, trophoblasts are exposed to a broad range of mechanical, oxidative, inflammatory, and hypoxic stimuli, each of which can independently or synergistically influence placental gene expression and cellular behavior. Therefore, while the fluid shear stress-based in vitro model provides significant mechanobiological insights, it only partially replicates the complex microenvironment of the preeclamptic placenta. Future investigations should integrate multiple pathophysiological stressors—including hypoxia and inflammatory mediators—to achieve a more physiologically comprehensive understanding of trophoblast dysfunction in PE.”

Thank you for all your valuable comments. We think the editors’ comments have significantly improved the quality of our manuscript by guiding our insights appropriately.

Min-A Kim, MD, PhD

Reviewer 2 Report

Comments and Suggestions for Authors

Lee et al. investigated novel predictive biomarkers for Preeclampsia (PE) in second-trimester amniotic fluid, a key strength that maximizes the predictive potential of the identified candidates by utilizing samples collected prior to symptom onset. This early collection time significantly enhances the clinical relevance for risk stratification. However, the initial discovery phase suffers from a significant methodological limitation: the reliance on RNA sequencing from a small cohort of only 14 total samples (n=7 PE, n=7 controls). Although HOOK2 displayed a large effect size, this limited sample size elevates the risk of false-positive differential expression and may yield results specific only to this small group. Consequently, these findings necessitate external validation in a larger, independent cohort. Furthermore, the functional validation utilizing the immortalized HTR-8/SVneo cell line does not perfectly recapitulate the behavior of primary human trophoblasts in vivo. Future research must prioritize validating the mechanistic role of HOOK2 in primary trophoblast cells to ensure translational relevance. Finally, while HOOK2 was the primary focus, the study identified 19 differentially expressed genes (DEGs), including three other validated candidates (CCDC160, CKB, and PARP15). The specific regulatory roles or interdependencies of these additional biomarkers remain largely undiscussed, representing a gap in the comprehensive understanding of the transcriptomic signature.

Author Response

Response to Reviewers

Reviewer #2

Comments 1: Lee et al. investigated novel predictive biomarkers for Preeclampsia (PE) in second-trimester amniotic fluid, a key strength that maximizes the predictive potential of the identified candidates by utilizing samples collected prior to symptom onset. This early collection time significantly enhances the clinical relevance for risk stratification. This early collection time significantly enhances the clinical relevance for risk stratification. However, the initial discovery phase suffers from a significant methodological limitation: the reliance on RNA sequencing from a small cohort of only 14 total samples (n=7 PE, n=7 controls). Although HOOK2 displayed a large effect size, this limited sample size elevates the risk of false-positive differential expression and may yield results specific only to this small group. Consequently, these findings necessitate external validation in a larger, independent cohort.

Response 1: We sincerely appreciate the reviewer’s constructive comment and fully agree that validation in a larger cohort is essential. Our study represents an initial discovery phase using AFS prospectively collected before the clinical onset of PE. Because PE occurs in only 2–8% of pregnancies, assembling amniotic fluid samples that later correspond to PE outcomes requires extensive longitudinal follow-up of a large screened population. Although the cohort size is modest, its prospective design and the rarity of second-trimester amniotic fluid linked to subsequent PE strengthen the dataset. We have explicitly acknowledged this as a major limitation in the revised manuscript. In line with the reviewer’s suggestion, we have carefully re-examined our presentation, and ongoing efforts to expand the sample collection and perform validation in an independent, larger cohort are underway. Based on the reviewer’s comment, we have added the following sentence to the limitations section of the Discussion:

“First, the small sample size limits statistical power and generalizability. Nonetheless, these prospectively collected second-trimester amniotic fluid samples—obtained before the clinical onset of PE—represent a rare and valuable resource for early biomarker discovery. Given the low prevalence of PE, future studies should validate these candidate biomarkers in larger, independent cohorts and across diverse biospecimens (e.g., maternal plasma or placental tissue) to enhance generalizability and translational relevance.”

Comments 2: Furthermore, the functional validation utilizing the immortalized HTR-8/SVneo cell line does not perfectly recapitulate the behavior of primary human trophoblasts in vivo. Future research must prioritize validating the mechanistic role of HOOK2 in primary trophoblast cells to ensure translational relevance.

Response 2: We thank the reviewer for this insightful comment. We fully agree that the immortalized HTR-8/SVneo cell line cannot completely reproduce the physiological characteristics of primary trophoblasts. Although this model is widely used for studying trophoblast invasion and mechanotransduction due to its stable phenotype and experimental reproducibility, it lacks the complex differentiation capacity and epigenetic landscape of primary cytotrophoblasts and syncytiotrophoblasts. Accordingly, we have added a statement in the limitations section acknowledging this constraint and emphasizing the need for future studies using primary trophoblasts or placental explant models to validate the mechanistic role of HOOK2 and confirm translational relevance as follows:

“Finally, the functional validation experiments were conducted using the immortalized HTR-8/SVneo trophoblast cell line. While this cell line is widely employed for investigating trophoblast migration, invasion, and response to shear stress because of its reproducibility and ease of culture, it may not fully reflect the molecular diversity or invasive behavior of primary trophoblasts in vivo. Therefore, although our findings in HTR-8/SVneo cells provide valuable mechanistic insights, future studies employing primary trophoblasts, placental explant cultures, or trophoblast organoid models are warranted to confirm the role of HOOK2 under physiologically relevant conditions.”

Comments 3: Finally, while HOOK2 was the primary focus, the study identified 19 differentially expressed genes (DEGs), including three other validated candidates (CCDC160, CKB, and PARP15). The specific regulatory roles or interdependencies of these additional biomarkers remain largely undiscussed, representing a gap in the comprehensive understanding of the transcriptomic signature.

Response 3: We thank the reviewer for this valuable comment. We agree that our study primarily focused on HOOK2, and a detailed analysis of the additional DEGs (CCDC160, CKB, and PARP15) was not included. These genes may participate in complementary biological pathways associated with cytoskeletal organization, cellular metabolism, and stress response, which are all relevant to trophoblast function and placental development. We have now expanded the discussion, as recommended by the reviewer, to acknowledge their potential biological significance and to highlight the need for integrative network-based analyses to determine how these DEGs interact within shared molecular pathways contributing to PE.

“Beyond HOOK2, our transcriptomic analysis also identified 19 DEGs, including CCDC160, CKB, and PARP15, which were also validated at the transcriptional level. However, the functional interconnections and biological significance of these genes in PE remain insufficiently characterized, warranting further investigation. Among these additional candidates detected in this study, CCDC160 encodes a coiled-coil domain–containing protein potentially linked to cytoskeletal organization and vesicular trafficking, pathways also relevant to trophoblast motility and endosomal regulation. CKB participates in intracellular ATP buffering and metabolic homeostasis and has been implicated in oxidative-stress–induced trophoblastic dysfunction and aberrant energy metabolism in PE placentas. PARP15, a poly (ADP-ribose) polymerase family member, is involved in DNA repair and post-translational regulation of stress responses, and this family has been increasingly associated with trophoblast inflammation and ferroptotic signaling in PE. Future network-based and functional analyses are needed to determine whether these DEGs act in coordinated pathways influencing trophoblast function and PE pathogenesis.”

Thank you for all your valuable comments. We think the editors’ comments have significantly improved the quality of our manuscript by guiding our insights appropriately.

Min-A Kim, MD, PhD

Round 2

Reviewer 2 Report

Comments and Suggestions for Authors

Accept after minor revision (corrections to minor methodological errors and text editing)

Author Response

Comments 1: Accept after minor revision (corrections to minor methodological errors and text editing)

Response 1: 

We greatly appreciate the reviewer’s recommendation of “Accept after minor revision (corrections to minor methodological errors and text editing)”. Following this suggestion, we have carefully reviewed and refined the Methods section to improve clarity and accuracy, ensuring consistency in terminology, normalization methods, and statistical descriptions. In addition, minor typographical and stylistic edits were made throughout the manuscript to enhance readability.

We have uploaded the revised version of the manuscript.

Thank you again for your time and consideration. We sincerely appreciate the reviewer’s positive feedback and the opportunity to further improve our manuscript.

Kind regards,
Min A Kim, MD, PhD